# Antibody DomainBed: Out-of-Distribution Generalization in Therapeutic Protein Design

## Abstract

Recently, there has been an increased interest in accelerating drug design with machine learning (ML). Active ML-guided design of biological sequences with favorable properties involves multiple design cycles in which (1) candidate sequences are proposed, (2) a subset of the candidates is selected using ML surrogate models trained to predict target properties of interest, and (3) sequences are experimentally validated. The returned experimental results from one cycle provide valuable feedback for the next one, but the modifications they inspire in the candidate proposals or experimental protocol can lead to distribution shifts that impair the performance of surrogate models in the upcoming cycle. For the surrogate models to achieve consistent performance across cycles, we must explicitly account for the distribution shifts in their training. We apply domain generalization (DG) methods to develop robust classifiers for predicting properties of therapeutic antibodies. We adapt a recent benchmark of DG algorithms, "DomainBed," to deploy DG algorithms across 5 domains, or design cycles. Our results suggest that foundational models and ensembling (in both output and weight space) lead to better predictive performance on out-of-distribution domains. We publicly release our codebase and the associated dataset of antibody-antigen binding that emulates distribution shifts across design cycles.

## 1 Introduction

A model trained to minimize training error is incentivized to absorb all the correlations found in the training data. In many cases, however, the training data are not sampled independently from the same distribution as the test data and such a model may produce catastrophic failures outside the training domain (Torralba & Efros, 2011; Zech et al., 2018; Beery et al., 2019; Koh et al., 2021b; Neuhaus et al., 2022). The literature on domain generalization (DG) aims to build a robust predictor that will generalize to an unseen test domain. A popular approach in DG extracts a notion of domain **invariance** from datasets spanning multiple training domains (Blanchard et al., 2011; Muandet et al., 2013; Arjovsky et al., 2019). This substantial body of work inspired by causality views the problem of DG as isolating the causal factors of variation, stable across domains, from spurious ones, which may change from training to test domains (Arjovsky et al., 2019; Ahuja et al., 2021; Rame et al., 2022a).

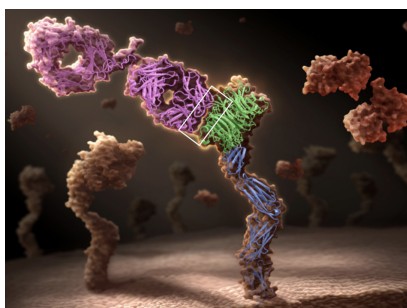

Figure 1: **Prediction task: antibody-antigen binding.** Antibody Onartuzumab [1] (pink) binds to MET (green and blue), a lung cancer antigen target, on the cell surface. The strength of the binding is determined by the binding site of the antibody interacting with the antigen, boxed in white.

Benchmarking efforts for DG algorithms, to date, have been largely limited to image classification tasks (e.g., Gulrajani & Lopez-Paz, 2020; Lynch et al., 2023). To prepare these algorithms for critical applications such as healthcare and medicine, we must validate and stress-test them on a wide variety of real-world datasets carrying selection biases, confounding factors, and other domain-specific idiosyncrasies. In this paper, we apply for the first time, DG algorithms to the problem of active drug design, a setting riddled with complex distribution shifts.

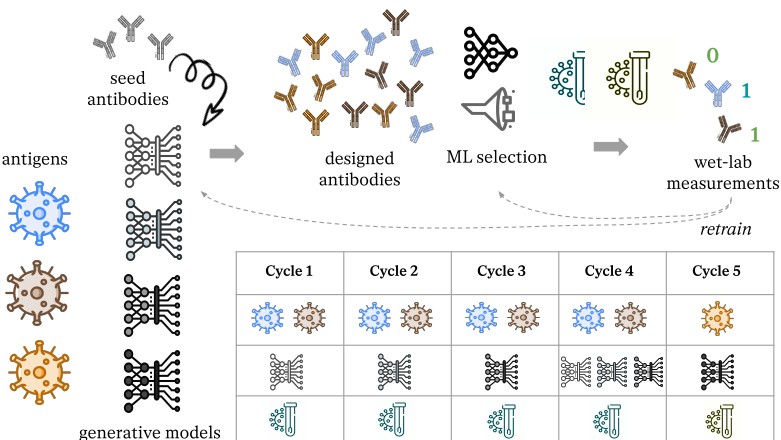

Figure 2: Active ML-guided design of antibodies effective against given antigens of interest typically proceeds by (1) developing multiple generative models that produce novel antibody designs given some starting seed antibody, (2) selecting the most promising designs using predictive models, and (3) experimentally validating the selected designs in the wet lab, and (4) updating the computational models with the measurements for the next cycle. In each cycle, the makeup of the targets, generative models, and/or experimental assays may vary.

The specific application we consider is that of characterizing the **binding affinity** of therapeutic antibodies. Antibodies are proteins used by the immune system to recognize harmful foreign substances (antigens) such as bacteria and viruses (Singh et al., 2018). They bind, or attach, to antigens in order to mediate an immune response against them. The strength of binding is determined by the binding site of the antibody (paratope) interacting with the antigen epitope (Figure 1). Antibodies that bind tightly to a given target antigen are highly desirable as therapeutic candidates.

The wet-lab experiments that measure the binding affinity of antibodies are costly and time-consuming. In active antibody design, we thus assign a surrogate ML model to predict binding and select the most promising candidates for wet-lab evaluation based on the predictions. Developing an accurate surrogate model is a challenging task in itself, because, as explained in more detail in section 2, the model may latch onto **non-mechanistic** factors of variation in the data that do not cause binding: identity of the target antigen, assay used to measure binding, distinguishing features of the generative models (either human experts or ML) that proposed the antibody, and "batch effects" that create heteroscedastic measurement errors.

We approach active drug design from the DG perspective. Active drug design, executed in multiple design cycles, informs the DG algorithm development, as it abounds in distribution shifts previously underexplored in the DG literature. Conversely, it benefits from a robust (surrogate) binding predictor. To summarize, this joint venture enables (1) impactful real-world benchmarking of DG algorithms and (2) development of robust predictors to serve active antibody design. Our contributions are the following:

- We open source a new antibody dataset for active drug design.
- We review and evaluate the latest DG algorithms in the context of drug design. Our work is the first large-scale benchmark study on large molecules, to our knowledge.
- We present some guidelines for best practices and highlight open questions for the field.

## 2 ACCELERATING ANTIBODY DESIGN WITH ML

**Problem formulation** Antibody design typically focuses on designing the variable region of an antibody, which consists of two chains of amino acids, called heavy and light chains. Each chain can be represented as a sequence of characters from an alphabet of 20 characters (for 20 possible amino acids). The heavy and light chains combined span $L \sim 290$ amino acids on average. We denote the sequences as $\boldsymbol{x} = (a_1, \ldots, a_L)$, where $a_l \in \{1, \ldots, 20\}$ corresponds to the amino acid type at

position $l \in [L]$. We experimentally measure the binding affinity $z \in \mathbb{R}$ from each sequence. For simplicity, we create a classification task by creating a binary label $y \in \{0, 1\}$ from $z$. We set $y = 1$ if $z$ exceeds a chosen minimum affinity value that would qualify as binding and $y = 0$ otherwise. Each antibody $\boldsymbol{x}_i$, indexed $i$, carries a label $y_i$ in one of the design rounds $r$, where $r \in \{0, \ldots, 4\}$. The labeled dataset for a round $r$ is a set of $n_r$ ordered pairs: $\mathcal{D}_r = \{(\boldsymbol{x}_i^r, y_i^r)\}_{i=1}^{n_r}$.

**Lab in the loop** Our antibody binding dataset is generated from an active ML-guided design process involving multiple design cycles, or *rounds*. As illustrated in Figure 2, each round consists of the following steps: **1**. Millions of candidate sequences are sampled from a suite of generative models, including variational autoencoders (Gligorijević et al., 2021; Berenberg et al., 2022), energy-based models (Tagasovska et al., 2022; Frey et al., 2023a) and diffusion models (Gruver et al., 2023; Frey et al., 2023b). **2**. A small subset of several hundred promising candidates is selected based on binding predictions from a surrogate binding classifier (Park et al., 2022). **3**. The wet lab experimentally measures binding. **4**. All models (generative and discriminative) are updated with new measurements. In Step 4, both the generative model and the surrogate classifier $\hat{f}_\theta$ are updated. Beyond being refit on the new data returned from the lab, the generative models may undergo more fundamental modifications in their architectures, pretrained weights, and training/regularization schemes.

A standard approach to supervised learning tasks is empirical risk minimization (ERM) (Vapnik, 1992). Let us first define the risk in each round $r$ as

$$\mathcal{R}^r(\theta) = \mathbb{E}_{(X^r, Y^r) \sim \mathcal{D}_{r_j}} \ell\left(\hat{f}_\theta(X^r), Y^r\right), \tag{1}$$

where $\ell$ is the loss function. ERM simply minimizes the training error, i.e., the average risk across all the training examples from all the rounds.

$$\mathcal{R}_{\text{ERM}}(\theta) = \mathbb{E}_{(X^r, Y^r) \sim \bigcup_{j \in [5]} \mathcal{D}_{r_j}} \ell\left(\hat{f}_\theta(X^r), Y^r\right) = \mathbb{E}_{r \sim p_{\text{train}}(r)} \mathcal{R}^r(\theta), \tag{2}$$

where $p_{\text{train}}(r)$ denotes distribution of the rounds in the training set. When we trained our surrogate classifier by ERM, it did not improve significantly even as the training set size increased over design rounds. In each subsequent round, representing the test domain, we observed that the classifier performance was close to random.

## 3 DOMAIN GENERALIZATION

The new measurements from the wet lab inspire modifications in the candidate proposals or experimental protocol, which lead to (feedback) covariate shift. DG has recently gained traction in the ML community as concerns about productionalizing ML models in unseen test environments have emerged (Rosenfeld et al., 2021). The interest in achieving out-of-distribution (OOD) generalization has spawned a large body of work in DG, which can be organized into the following families of approaches:

**DG by invariance** This paradigm has mainly been motivated by learning "causal representations." Invariant causal prediction (Peters et al., 2016) frames prediction in the language of causality, assuming that the data are generated according to a structural equation model (SEM) relating variables in a dataset to their parents by a set of mechanisms, or structural equations. The major assumption of ICP is the partitioning of the data into environments $e \in E$ such that *each environment corresponds to interventions on the SEM*, but importantly, the mechanism by which the target variable is generated via its direct parents is unaffected (Pearl, 2009). This means that the true causal mechanism of the target variable is fixed, while other features of the generative distribution can vary. This motivates the objective of learning mechanisms that are stable (invariant) across environments with the hope that they would generalize under unseen, valid [2] interventions.

The ultimate goal of these frameworks is to learn an "optimal invariant predictor" which uses only the invariant features of the SEM. We assume that high-dimensional observations take lower-dimensional representations governed by a generative model. In the invariant learning paradigm, it is common to define the task as learning invariant representations of the data, rather than seeking

---

[2]Interventions are considered valid if they do not change the structural equation of $Y$.

invariant features in the observation space. This problem setup has inspired a plethora of algorithms, starting with IRM (Arjovsky et al., 2019)

$$\mathcal{R}_{\text{IRM}} = \min_{\substack{\Phi:\mathcal{X}\to\mathcal{H}; \\ w:\mathcal{H}\to\mathcal{Y}}} \sum_{e\in E_{tr}} \mathcal{R}^e(w\cdot\Phi) \; s.t. \; w \in \argmin_{\bar{w}:H\to Y} \mathcal{R}^e(\bar{w}\cdot\Phi) \; \forall e \in E.$$

IRM assumes invariance of $\mathbb{E}[y|\Phi(x)]$—that is, invariance of the feature-conditioned label distribution. Follow-up studies make a stronger assumption on invariance based on higher-order conditional moments (Krueger et al., 2021; Xie et al., 2020). Though this perspective has gained traction in the last few years, it is somewhat similar to methods from domain adaptation, such as DANN (Ganin et al., 2016) and CORAL (Sun & Saenko, 2016), which minimize domain shift by aligning the source and target feature distributions. Another line of work considers learning shared mechanisms by imposing invariance of the gradients distributions across domains. In our setup the gradients per environment are:

$$\mathbf{g}_e = \mathbb{E}_{(X^e,Y^e)\sim\mathcal{D}_e}\nabla_\theta\ell\left(\hat{f}_\theta(X^e), Y^e\right),$$

Parascandolo et al. (2020) initiated such approaches, aiming to learn invariant explanations by replacing the arithmetic mean in gradient descent with a geometric one, hence promoting agreements of the gradients across domains. Other popular gradient based approaches include Fish (Shi et al., 2021) which match the first moments of the gradient distributions, and Fishr (Rame et al., 2022a) which similarly to CORAL matches the variance in gradient space.

**DG by ensembling**  We consider two types of ensembling strategies that do not use domain information (i.e., environment labels). First, *output-space ensembles* combine multiple independently trained models for an input $\mathbf{x}$ as follows:

$$\arg\max_k \; \text{Softmax}\left(\frac{1}{M}\sum_{m=1}^{M} f(\mathbf{x};\theta_m)\right)_k$$

where $M$ is the total number of models in the ensemble, $\theta_m$ are the parameters of the $m$-th model, and the sub-script $(\cdot)_k$ denotes the $k$-th element of the multiclass vector argument. A standard ensembling approach, deep ensemble, combines models trained with different initializations and was shown to achieve strong robustness to OOD data (Lakshminarayanan et al., 2017).

Second, *weight-space ensembles*. Given $M$ individual member weights $\{\theta_m\}_{m=1}^{M}$ corresponding to individual models, Weight averaging (WA), is defined as:

$$f_{WA} = f(\cdot, \theta_{WA}), \text{where} \quad \theta_{WA} = \frac{1}{M}\sum_{m=1}^{M}\theta_m$$

A combination of different weight averaging and fine tuning resulted in different methods e.g. Stochastic Weight Average (Izmailov et al., 2018), Simple Moving Average (Arpit et al., 2022), Diverse Weight Averaging a.k.a model soup (Wortsman et al., 2022; Rame et al., 2022b). These models usually leverage pre-trained *foundational* models (Bommasani et al., 2021).

### 3.1 HYPOTHESIS - INVARIANT FEATURE REPRESENTATIONS OF ANTIBODIES

Our lab-in-the-loop (section 2) offers a unique testbed for DG algorithms. In particular, we attempt to answer the question:

*Can DG algorithms help in developing robust predictors for antibody design? Do learnt invariant representations align with the physics-based features causing binding properties?*

We propose to consider the design rounds $r \in \{0, \dots, 4\}$ as environments $e$, since rounds do correspond to valid interventions — our design cycles should not impact the true causal mechanism governing binding affinity. There are two types of features that a binding classifier can learn:

- *Invariant (causal) features*: various physico-chemical and geometric properties at the interface of antibody-antigen binding (Figure 1) and
- *Spurious correlations*: Other round-specific features that are byproducts of different folding algorithms, generative models, measurement assay types, antigen targets, etc.

We expect DG algorithms to be able to distinguish between the two, and only make use of the features invariant across rounds in their predictions.

## 4 RELATED WORK

Existing benchmarks for investigating OOD generalization are mostly image-based (e.g., Gulrajani & Lopez-Paz, 2020; Koh et al., 2021a). Yet it is unclear if the conclusions for these results transfer to other real-world applications. Our benchmark aims at answering this question, focusing on the topic of drug discovery.

In the drug discovery setting, some benchmarks have been proposed, however, they are restricted to small molecules, or compounds comprised of 20-100 atoms and typically weighing less than 1,000 daltons. Ji et al. (2023) explores shifts due to assay types, scaffolds, and molecular sizes. Tossou et al. (2023) studies data splitting strategies in two deployment settings: virtual screening and *de novo* generation.

We propose a benchmark for therapeutic proteins including distribution shifts likely to occur in active ML-guided design. Proteins fall under large molecules, made of *thousands* of atoms. Being much larger than small molecules, they are arguably more complex and more challenging to characterize. In particular, the three-dimensional folded structure of the protein is highly indicative of its function. When predicting functional properties of proteins, working with structure-aware representations is key. Proteins present unique modeling challenges and, to our knowledge, this is the first large-scale OOD benchmark on large molecules.

## 5 ANTIBODY DOMAINBED

### 5.1 THERAPEUTIC PROTEIN DATASET

The main objective of this benchmark is to emulate, with high-fidelity, a real-world active drug design setup. To create a realistic, publicly accessible dataset, we propose the following procedure: (1) Collection of open data antibody-antigen complex structures, antibody wild types (seeds), and corresponding mutants; (2) Training generative models and sampling candidates with different properties (edit distances from training data, targets of interest, different initial complex structures); (3) Computing a proxy for binding from physics-based models for all designs from Step 2; and (4) Splitting the labeled dataset into a number of meaningful environments.

**Step 1: Data curation.** We rely on the latest version (at the time of writing of this manuscipt) of the popular *Structural Antibody Database*, SAbDab Dunbar et al. (2014); Raybould et al. (2020); Schneider et al. (2022) which catalogs 7,689 PDB structures, and a recent derivative *Graphinity*, Hummer et al. (2023) which extends SAbDab to a synthetic dataset of a much larger scale (nearly 1M) by introducing systematic point mutations in the CDR3 loops of the antibodies in the original SAbDab complexes. In this benchmark, we select the antibody wild types and mutants related to three popular antigens - HIV1 [3], SARS-CoV-2[4] and HER2 [5].

**Step 2: Sampling antibody candidates.** To emulate the active drug discovery pipeline, we need a suite of generative models for sampling new candidate designs for therapeutic antibodies. We run the *Walk Jump Sampler* (WJS; Frey et al., 2023b), a method building on the neural empirical Bayes framework (Saremi & Hyvarinen, 2019). WJS separately trains score- and energy-based models to learn noisy data distributions and sample discrete data. The energy-based model is trained on noisy samples, which means that by training with different noise levels $\sigma$, we obtain different generative models. Higher $\sigma$ corresponds to greater diversity in the samples and higher distances from the starting seed. We used four values for the noise parameter, namely $\sigma \in [0.5, 1.0, 1.5, 2.0]$.

**Step 3: Labeling candidates.** Since wet-lab assays to experimentally measure binding affinity are prohibitively expensive, we use computational frameworks which, by modeling changes

---

[3]The most common type of Human Immunodeficiency Virus that can lead to AIDS. HIV attacks the body's immune system by destroying CD4 cells, which help your body fight infections

[4]Severe acute respiratory syndrome coronavirus 2, is a strain of coronavirus that causes COVID-19, the respiratory illness responsible for the COVID-19 pandemic.

[5]Human epidermal growth factor receptor 2 is a gene that makes a protein found on the surface of all breast cells. Breast cancer cells with higher than normal levels of HER2 are called HER2-positive which signals breast cancer may grow quickly and possibly come back.

Table 1: Dataset overview. Number of samples in parentheses.

| Environment | Antigens | Generative Model | Annotation | Description - emulating env with: |
|---|---|---|---|---|
| **env 0** | HIV1 (186) SARS-CoV-2 (1117) | WJS $\sigma \in [0.5]$ | $\Delta\Delta G$ computed with pyRosetta | distribution shifts due to generative model (covariate shift) |
| **env 1** | HIV1 (780) SARS-CoV-2 (3096) | WJS $\sigma \in [1.0, 1.5]$ | $\Delta\Delta G$ computed with pyRosetta | distribution shifts due to generative model (covariate shift) |
| **env 2** | HIV1 (275) SARS-CoV-2 (1469) | WJS $\sigma \in [2.0]$ | $\Delta\Delta G$ computed with pyRosetta | distribution shifts due to generative model (covariate shift) |
| **env 3** | HIV1 (552) SARS-CoV-2 (3142) | WJS $\sigma \in [0.5, 1.0, 1.5, 2.0]$ | $\Delta\Delta G$ computed with pyRosetta | distribution shifts due to generative model (covariate shift) |
| **env 4\*** | HER2 (2471) | point mutations in CDR3 | $\Delta\Delta G$ computed with FoldX | (i) zero-shot generalization to new target (*de novo* design) (ii) concept drift |
| **env 5\*** | HER2 (226) | internally generated | $K_d$ experimentally measured with SPR | verifying generalizability to $K_d$ measurements. |

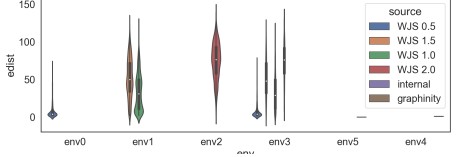 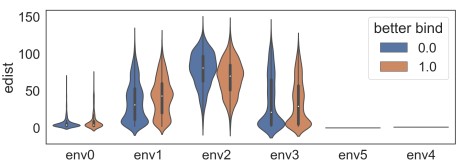

Figure 3: Antibody Domainbed environments. Left - edit distance a.k.a sequence similarity between designs and seeds. Right - binding properties per generative model.

in binding free energies upon mutation (interface $\Delta\Delta G = \Delta G_{\text{wild type}} - \Delta G_{\text{mutant}}$), allow for large-scale prediction and perturbation of protein–protein interactions Barlow et al. (2018). We use the `pyrosetta` (pyR) Chaudhury et al. (2010) implementation of the `InterfaceAnalyzerMover`, namely the scoring function `ref2015`, to compute the difference in energy prior and after mutations in a given antibody-antigen complex. After removing highly uncertain labels between -0.1 and 0.1 kcal/mol (Sirin et al., 2016; Hummer et al., 2023), we attach binary labels to each candidate of the generative models: label 1 if $\Delta\Delta G < -0.1$ (stabilizing) and label 0 if $\Delta\Delta G > 0.1$ (destabilizing). While the computed $\Delta\Delta G$ represent weak proxies of binding free energy, they have been shown to be predictive of experimental binding (Mahajan et al., 2022; Hummer et al., 2023). See B.1 for details.

**Step 4: Splitting into environments.** To emulate the active drug design setup where new sources of distribution shift might appear (e.g., new generative models, antigen targets, experimental assays) at each iteration, we split the overall dataset into five cycles. Each cycle corresponds to a different domain or environment in the language of domain generalization algorithms. In Table 1 we summarize how the overall data pool was split into environments, as well as some summary statistics of each environment. This dataset split mimics sub-population shifts due to the generative model, which produces antibody sequences with different edit distances from the seed sequences. The WJS model with $\sigma$=0.5 ($\sigma$=2.0) produces antibody designs close to (far from) the seed. Environment 4 has been partially included in the experiments because it introduces severe distribution shift in the form of concept drift and label shift, as it represents a completely new target and a different labeling mechanism than the rest. We report some preliminary results in this extreme setup in subsection A.3.

## 5.2 PROTEIN SEQUENCE ARCHITECTURES

Different DG solutions assume different types of invariance, and propose algorithms to estimate them from data. DomainBed (Gulrajani & Lopez-Paz, 2020) is a benchmark suite that contains the majority of DG algorithms developed in the past two years and a benchmark environment that compares them across multiple natural image datasets.

To adapt DomainBed our antibody design context, we modify its featurizer to accept biological sequences as input. We do so by (i) implementing a preprocessing module to align the antibody sequences (using the AHo numbering scheme suitable for antibody variable domains; Honegger & PlùÈckthun, 2001) and (ii) replacing the default ResNet (He et al., 2016) with one of the following more suitable architectures:

**SeqCNN:** Our SeqCNN model consists of an embedding layer with output dimension 64, and two consecutive convolutional layers with kernel sizes 10 and 5 respectively, stride of 2 followed by

Figure 4: MMD in the learned features of ESM between every pair of environments. DG algorithms result in features that are significantly more uniform across environments.

ReLU nonlinearities. The CNN output is pooled with a mixing layer of size 256. This identical architecture is applied for both the antibody and the antigen sequence before concatenating them and passing them to the classification head.

**Finetuned ESM2:** We finetune the 8M-parameter ESM2 Lin et al. (2023), a protein language model pretrained on experimental and high-quality predicted structures of general proteins. For speed, we used a single ESM2 model for the two antibody chains as well as the antigen. One potential challenge with fine-tuning a single ESM2 model on three protein chains is that they are OOD for ESM2, which was pretrained on single chains. To address this, we follow the tricks used in ESMFold (Lin et al., 2023): (1) adding a 25-residue poly-glycine linker between the heavy and light antibody chains and between the antibody and antigen and (2) implementing a jump in residue index into the positional embeddings at the start of each new chain. The tricks significantly boosted the classifier performance, signaling that the structure information in the ESM embeddings was important.

**GearNet (Zhang et al., 2022):** We finetune GearNet-Edge MVC , a general-purpose structure-based protein encoder pre-trained with a contrastive learning objective on AlphaFold2 predictions. Notably, the pre-training dataset does not include complexes or significant antibody structures. As a result, the Antibody DomainBed environments are OOD for GearNet-ESM. We make no adjustments, using the default structure graph construction and featurisation.

Additionally, we extend DomainBed to include the moving average ensembling as in (Arpit et al., 2022) as well as functional ensembling, or stochastic weight averaging (Izmailov et al., 2018) for all DG baselines in the repository. We denote the averaged solutions with *-ENS* suffix.

**Open source** We open-source our efforts so that other researchers can continue further evaluations on similar biological datasets. With this paper, we make publicly available the "Antibody DomainBed," a codebase aligned with the DomainBed suite, here. We also release a public benchmark antibody dataset available here.

## 6 BENCHMARKING RESULTS AND ANALYSIS

**Models** We tested vanilla ERM against three classes of DG methods. These are methods that leverage the domain information to enforce invariance on representations (CORAL; Sun & Saenko, 2016), on the predictor (IRM; Arjovsky et al., 2019), or on gradients (Fish; Shi et al., 2021). We additionally test ensembling techniques, namely deep ensembles, (simple moving average or SMA; Arpit et al., 2022), that do not exploit domain information but combine different models to reduce the effect of covariate shift (Rame et al., 2022b). The last class of models has achieved state-of-the-art results on image datasets (Cha et al., 2021; Arpit et al., 2022; Rame et al., 2022b). We perform early stopping for the members of the deep ensembles in the training-domain validation setting as in Arpit et al. (2022). The best model is ensembled based on validation performance over each trial. From the available DG algorithms in DomainBed, this corresponds to 6 algorithm baselines with 5 hyperparameter configurations (with varying batch size, weight decay, and learning rate) for ESM-based architectures and 20 for SeqCNN-based architectures. As we perform 3 seed repetitions for each configuration, we have a total of 1,890 experiments. We report the results from *model selection method: training domain validation set*, which is a leave-one-environment-out model selection strategy. We (1) split the data into train and test environments, (2) pool the validation sets of each training domain to create an overall validation set, and (3) choose the model maximizing the accuracy (minimizing the negative log-likelihood) on the pooled validation set.

**Metrics and evaluation** We evaluate the performance in terms of accuracy. As Gulrajani & Lopez-Paz (2020), we test two types of validation: training-domain and test-domain in subsection 6.1. Then, we evaluate the saliency maps produced by representative domain-invariant and ensembling models. In Appendix, we repeat the analysis for a second version (a new split) of the dataset containing an environment with antibody designs of a new antigen.

## 6.1 DISCUSSION

Based on the results in Table 2 and Table 3 we observe that, among the different DG paradigms, ensembling-based models work the best and functional ensembles improve all base algorithms. Additionally, as the model size increases (from SeqCNN to ESM2), the improvement is more pronounced. When the validation set is not aligned with the test set, there is no significant difference between models in terms of performance. The only class of models robust to such a setup is again, functional ensembles; they seem to work well even if the validation set is not aligned to the test set.

| FW1 | CDR1 | FW2 | CDR2 | FW3a | FW3b | CDR3 | FW4 |

Figure 5: Segments of a single antibody chain (heavy or light). "FW" denotes "framework."

SMA is the only DG model that has advantage over ERM across all domains. From the MMD plots in Figure 4, we do notice that the invariance based models indeed learn more uniform representations across domains. It seems, however, that invariant representations do not necessary translate to better predictive performance.

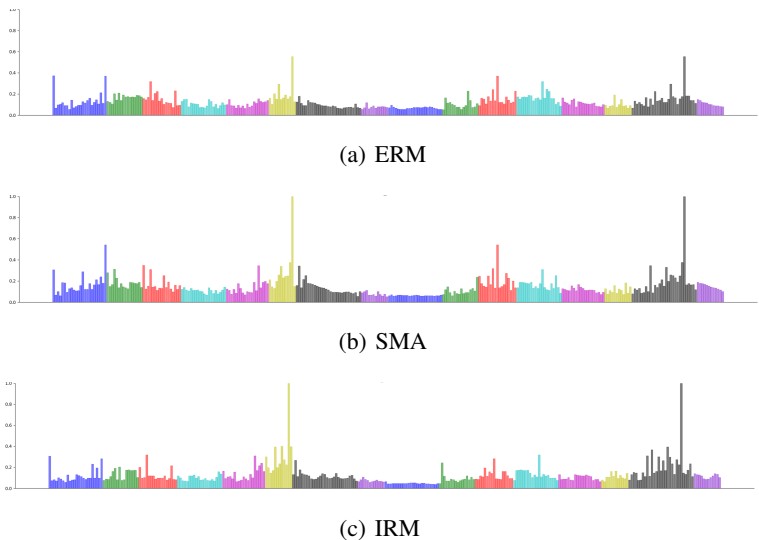

(a) ERM

(b) SMA

(c) IRM

Figure 6: Saliency visualizations on the antibody amino-acid positions for heavy and light chains, after aligning to uniform lengths as in (Honegger & PluÈckthun, 2001). x-axis represents the 298 positions after alignment. Colors of bars represent functional segments. Relative to ERM-SMA and IRM, ERM displays muted behavior in the regions known to interact with the antigen paratope.

Figure 6 shows the gradient sensitivity (Baehrens et al., 2010; Simonyan et al., 2013) with respect to the antibody residue positions. The different colors represent functional segments, illustrated in Figure 5. Details on the organisation and role of the different segments in an Antibody is included in Appendix C. Spikes can be interpreted as positions that strongly affected the prediction. Whereas ERM saliency is muted in the right edges of framework 3b (yellow) in the heavy chain and in CDR 3 (gray) in the light chain, ERM-SMA and IRM is activated in these positions. Heavy framework 3b, adjacent to heavy CDR 3, is heavily implicated in antigen binding (Morea et al., 1998). Light CDR 3 displays high diversity, though to a lesser extent than the heavy counterpart. Light CDR 3 is known to assume one of the few canonical structures that determines antigen recognition (Teplyakov & Gilliland, 2014), which may explain the models' sensitivity to this segment. Our main takeaways for robust prediction in therapeutic protein design are as follows:

Table 2: Model selection: training-domain validation set

| Algorithm | Env 0 | Env 1 | Env 2 | Env 3 | Avg |
|---|---|---|---|---|---|
| random | $50.5 \pm 0.0$ | $52.4 \pm 5.2$ | $38.8 \pm 0.0$ | $49.1 \pm 1.1$ | 47.7 |
| **ESM** | | | | | |
| ERM | $58.7 \pm 0.7$ | $66.8 \pm 0.4$ | $69.0 \pm 0.5$ | $65.9 \pm 0.2$ | 65.1 |
| ERM-ENS | 62.9 | 69.7 | 71.6 | 67.9 | 68.0 |
| SMA | $59.6 \pm 1.1$ | $66.7 \pm 0.0$ | $70.1 \pm 0.1$ | $66.2 \pm 0.3$ | 65.7 |
| SMA-ENS | 61.5 | 68.1 | 73.4 | 67.7 | 67.7 |
| IRM | $60.0 \pm 0.9$ | $64.4 \pm 0.2$ | $69.6 \pm 0.9$ | $63.5 \pm 0.6$ | 64.4 |
| IRM-ENS | 62.7 | 69.1 | 71.3 | 67.0 | 67.5 |
| CORAL | $60.0 \pm 0.4$ | $66.9 \pm 0.2$ | $69.6 \pm 0.5$ | $65.7 \pm 0.4$ | 65.5 |
| CORAL-ENS | 62.6 | 69.4 | 71.7 | 68.1 | 68.0 |
| VREx | $58.7 \pm 0.6$ | $66.1 \pm 0.9$ | $69.0 \pm 0.6$ | $65.7 \pm 0.1$ | 64.9 |
| VREx-ENS | 61.1 | 67.2 | 71.9 | 67.9 | 67.0 |
| Fish | $59.3 \pm 1.2$ | $66.2 \pm 0.7$ | $69.5 \pm 0.1$ | $66.4 \pm 0.4$ | 65.3 |
| **SeqCNN** | | | | | |
| ERM | $63.2 \pm 1.0$ | $66.2 \pm 0.9$ | $66.2 \pm 0.8$ | $64.9 \pm 0.1$ | 65.1 |
| ERM-ENS | 62.3 | 66.1 | 69.7 | 65.9 | 66.0 |
| SMA | $61.8 \pm 0.9$ | $66.5 \pm 0.3$ | $66.1 \pm 0.2$ | $64.9 \pm 0.3$ | 64.9 |
| SMA ENS | 58.6 | 66.9 | 68.8 | 66.1 | 65.0 |
| IRM | $60.0 \pm 0.9$ | $64.4 \pm 0.2$ | $69.6 \pm 0.9$ | $63.5 \pm 0.6$ | 64.4 |
| IRM-ENS | 62.4 | 66.5 | 73.2 | 65.1 | 66.8 |
| CORAL | $62.0 \pm 1.5$ | $66.2 \pm 0.3$ | $66.1 \pm 0.8$ | $64.0 \pm 0.5$ | 64.6 |
| CORAL-ENS | 60.9 | 67.1 | 68.9 | 65.2 | 65.5 |
| VREx | $60.1 \pm 1.6$ | $65.7 \pm 1.0$ | $66.3 \pm 0.6$ | $64.9 \pm 0.4$ | 64.2 |
| VREx-ENS | 61.5 | 66.9 | 68.2 | 66.1 | 65.7 |
| Fish | $62.0 \pm 0.9$ | $66.9 \pm 0.8$ | $68.6 \pm 0.6$ | $65.8 \pm 0.3$ | 65.8 |
| **GearNet** | | | | | |
| ERM | $60.4 \pm 0.6$ | $79.2 \pm 0.3$ | $85.3 \pm 0.5$ | $75.1 \pm 1.1$ | 75.0 |
| ERM-ENS | **64.9** | **82.0** | **88.0** | **78.4** | **78.3** |
| VREx | $58.3 \pm 0.4$ | $73.1 \pm 0.5$ | $82.1 \pm 1.3$ | $71.1 \pm 1.3$ | 71.2 |
| VREx-ENS | 59.5 | 77.6 | 86.4 | 73.3 | 74.2 |

- Leveraging foundational models help performance across all DG models.
- Choosing a good validation set is important.
- Ensembling helps all baseline models.
- Leveraging $\Delta\Delta G$ predictions is beneficial for better performance on binding prediction.
- Ensembled DG models provide robust predictions on unseen antigen targets, essential for (de-novo) drug discovery.

## 7 CONCLUSION

We publicly release the codebase of our Antibody DomainBed pipeline as well as the associated antibody dataset representing the first large-molecule OOD benchmark of its kind. Our experiments suggest that DG methods have the capacity to significantly aid protein property prediction in the presence of complex distribution shifts. Antibody DomainBed enables the exploration of key theoretical questions that, when addressed, would maximize the impact of DG methods on biological problems. One question is: how can we generate diverse environments that would lead to optimal performance at test time? Related to this is the question of how to choose the different configurations governing each environment in a manner that would maximize learning for each DG algorithm. Finally, the ultimate OOD quest in the context of antibody design would be to produce accurate predictions for a completely new antigen. This will require the DG models to pick up on truly causal, or mechanistic, features governing the binding interaction. By open-sourcing our code as well as the data, we hope to motivate other ML researchers to aim at addressing impactful real-world applications close to the production setting.

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

# A ADDITIONAL RESULTS FROM EXPERIMENTS

## A.1 MODEL SELECTION: TEST-DOMAIN VALIDATION SET

Table 3: Model selection: test-domain validation set, corresponding to the training domain validation set in the main text.

| Algorithm | Env 0 | Env 1 | Env 2 | Env 3 | Avg |
|---|---|---|---|---|---|
| random | $50.5 \pm 0.0$ | $52.4 \pm 5.2$ | $38.8 \pm 0.0$ | $49.1 \pm 1.1$ | 47.7 |
| **ESM** | | | | | |
| ERM | $60.5 \pm 0.9$ | $67.6 \pm 0.6$ | $69.3 \pm 0.5$ | $66.1 \pm 0.5$ | 65.9 |
| ERM-ENS | 62.1 | 69.1 | 73.2 | 69.2 | 68.4 |
| SMA | $62.9 \pm 0.8$ | $68.3 \pm 0.4$ | $70.7 \pm 0.5$ | $66.7 \pm 0.4$ | 67.1 |
| SMA-ENS | 63.8 | 70.0 | 72.8 | 68.3 | 68.7 |
| IRM | $59.2 \pm 0.5$ | $65.5 \pm 1.8$ | $68.6 \pm 0.3$ | $63.6 \pm 1.1$ | 64.2 |
| IRM-ENS | 62.3 | 68.9 | 70.4 | 66.9 | 67.1 |
| CORAL | $60.7 \pm 0.1$ | $67.5 \pm 0.1$ | $68.4 \pm 0.2$ | $66.8 \pm 0.3$ | 65.8 |
| CORAL-ENS | 62.6 | 70.5 | 71.9 | 67.7 | 68.2 |
| VREx | $60.4 \pm 1.1$ | $65.4 \pm 0.9$ | $69.6 \pm 0.5$ | $65.9 \pm 0.2$ | 65.3 |
| VREx-ENS | 62.4 | 68.3 | 73.0 | 68.1 | 68.0 |
| Fish | $61.0 \pm 1.6$ | $65.9 \pm 0.5$ | $70.6 \pm 1.3$ | $66.3 \pm 0.3$ | 66.2 |
| **SeqCNN** | | | | | |
| ERM | $61.4 \pm 1.4$ | $64.4 \pm 0.1$ | $66.5 \pm 0.6$ | $63.8 \pm 0.5$ | 64.0 |
| ERM-ENS | 62.6 | 66.2 | 68.0 | 66.5 | 65.8 |
| SMA | $57.0 \pm 0.4$ | $65.9 \pm 0.3$ | $65.8 \pm 0.2$ | $64.5 \pm 0.4$ | 63.3 |
| SMA-ENS | 61.2 | 67.0 | 67.1 | 66.2 | 65.4 |
| IRM | $61.7 \pm 1.4$ | $65.8 \pm 0.5$ | $68.6 \pm 1.6$ | $63.7 \pm 0.8$ | 64.9 |
| IRM-ENS | 62.70 | 64.34 | 63.51 | 71.56 | 65.5 |
| CORAL | $58.3 \pm 1.5$ | $65.3 \pm 0.5$ | $65.9 \pm 0.9$ | $63.2 \pm 0.8$ | 63.2 |
| CORAL-ENS | 62.0 | 67.4 | 68.4 | 65.4 | 65.8 |
| VREx | $58.4 \pm 1.4$ | $65.3 \pm 0.3$ | $65.1 \pm 0.3$ | $63.8 \pm 0.4$ | 63.2 |
| VREx-ENS | 60.9 | 66.8 | 67.83 | 66.73 | 65.6 |
| Fish | $59.9 \pm 2.6$ | $67.0 \pm 0.1$ | $69.4 \pm 0.4$ | $65.3 \pm 0.4$ | 65.4 |
| **GearNet** | | | | | |
| ERM | $62.7 \pm 1.6$ | $79.7 \pm 0.4$ | $85.9 \pm 1.2$ | $76.0 \pm 0.5$ | 76.1 |
| ERM-ENS | **64.3** | 81.7 | **88.6** | **78.1** | **78.2** |
| VREx | $58.0 \pm 0.1$ | $73.8 \pm 1.0$ | $82.3 \pm 1.5$ | $72.5 \pm 0.3$ | 71.6 |
| VREx-ENS | 62.2 | 77.3 | 86.9 | 74.6 | 75.3 |

## A.2 IRM V1 PENALTY

The IRM v1 is a relaxation of IRM originally proposed by Arjovsky et al. (2019), where the classifier $w$ is fixed to a scalar 1. The loss function includes a penalty term $||\nabla_{w|w=1} R_e(w \cdot \Phi)||^2$. Theorem 4 of Arjovsky et al. (2019) is used to justify the use of this term as the invariance penalty for all differentiable loss functions, such as the cross-entropy loss function used in this paper. Figure 7 plots this invariance penalty for various algorithms across the four environments. IRM carries the least IRM v1 penalty overall, as expected. Other invariance-based algorithms also carry low penalty values, while ERM and ERM-SMA have high penalty values, particularly in Env 0. The high relative penalty value in Env 0 is consistent with the accuracy for Env 0 being the lowest (see Table 2, Table 3).

### A.2.1 PLATFORM SPECIFICATIONS

All experiments have been executed on NVIDIA A100 Tensor Core GPU.

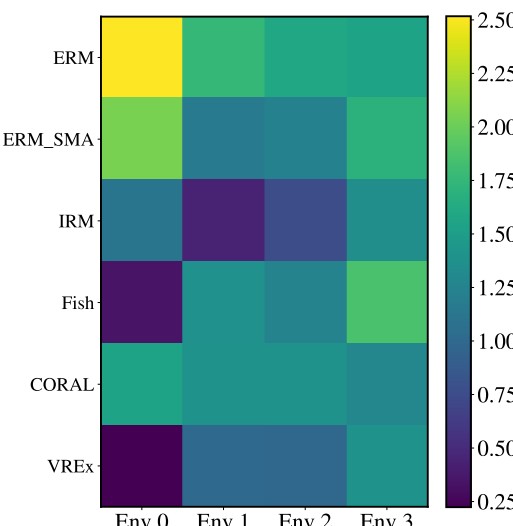

Figure 7: ERM has the highest IRM v1 penalty across all environments. The highest penalty value for ERM in Env 0 is consistent with the accuracy for Env 0 being the lowest (see Table 2, Table 3).

### A.3 GENERALIZING TO NEW ANTIGEN TARGETS

One challenging yet practical scenario is being able to predict a property of interest for a new antigen that has not been seen during training. We were thus motivated to evaluate the DG algorithms to an unseen target, HER2. The data in this new domain, consisting only of HER2 designs, was obtained from Graphinity (Hummer et al., 2023), which significantly differs from the WJS generative model in the distribution of antibody sequences. Graphinity it is a brute-force method that starting from a seed sequence, applies all possible point mutations at each of the positions in the CDR3 loop of an antibody. Which means that this method produces antibodies with edit distance of only 1 (one amino acid different from the starting seed). This difference in sequence distance amounts to a change in covariate and sub-population shift, compared to the other environments with WJS designs. Additionally, since these designs were scored using a different $\Delta\Delta G$ model, FoldX, this environment also includes label shift. With so much distribution shift compared to the rest of the data, it is expected that models trained on the other targets (HIV and SARS-Cov-2) will not generalize to HER2 designs. However, we wanted to investigate if the DG algorithms will have some advantage over vanilla ERM.

The table below summarizes the results where environments consist of the data curated as described in section section 5. The results unfortunately are not in favour of any of the DG algorithms. There is an advantage of the SeqCNN framework achieving higher accuracy on environment 4, however, those number are still around 50% and hence we can not consider them useful since we could not use such model in practice.

We are further investigating if including a new target but without label shift (i.e. using the pyRosetta scores) may deliver better results. As previously mentioned, having a model that can reliably predict binding or other molecular properties while being antigen agnostic is of crucial importance in accelerating drug discovery and design.

## B DATASET PROPERTIES

In this section we evaluate the validity of the antibodies in our synthetic library. We do so, to ensure quality and reliability of the proposed benchmark. We evaluate the following properties:

1. *naturalness* - measures the similarity of the antibody sequences in Antibody Domainbed to antibodies in OAS (Olsen et al., 2022), the largest publicly available database of natural

antibodies. These scores come from the perplexity of a language model trained on the OAS database (Melnyk et al., 2023). Higher is better.

2. *OASis -Percentile/Identity/Germline Content* we include three scores computed with the recent publicly available humanness evaluation platform BioPhi (Prihoda et al., 2022). We used the default setup in relaxed mode. Briefly, OASis identity score for an input sequence is calculated as the fraction of peptides with prevalence in therapeutic antibodies curated from OAS (Olsen et al., 2022). OASis Percentile converts the identity score to 0-100 range based on therapeutic antibodies such that 0% OASis percentile score corresponds to the least human and the 100% OASis percentile score corresponds to the most human antibody in the clinic. Germline content is yet another humanness score, which represents the percent sequence identity with a concatenation of the nearest human V and J gene (percent sequence identity with nearest human germline) with regards to IMGT Gene-DB (Giudicelli et al., 2005).

3. *bio-informatics properties* hydrophobicity - a measure of the degree of affinity between water and a side chain of an amino acid; pi charge - the pH at which the antibody has no net electrical charge, this value depends on the charged amino acids the antibody contains; and aromaticity - binding of the two complementary surfaces is mostly driven by aromatic residues. For these three properties we use the corresponding bioPython implementations, and we compare the range of values to in-vitro functional antibodies (env 5).

4. *diamond* we use this score to explore closeness of the proposed designs to the OAS database, by fast sequence alignment inspired by (Buchfink et al., 2021). Higher scores are preffered.

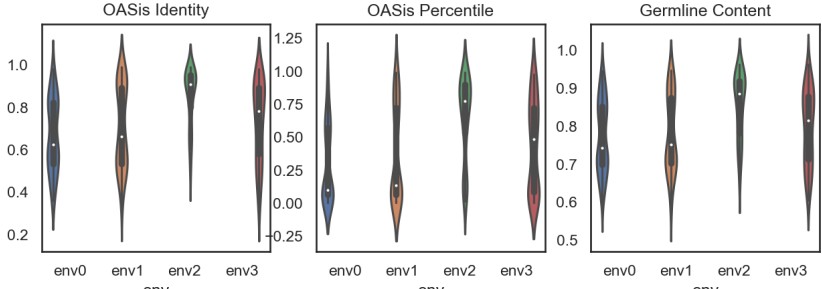

Figure 8: OASis properties for Antibody domainbed computed using BioPhi (Prihoda et al., 2022). Note that env 5 was omitted due to proprietary reasons.

Our results are presented in Figure 9 and Figure 8. In Figure 9, first row, naturalness and diamond score, we confirm that WJS generated antibodies (env 0-3) have properties close to observed antibodies, and even more so, they achieve better scores than the single point mutations in Graphinity (env4) and the human-expert designs from internal in-vitro experiments (env5). Next, in the second row of Figure 9, we notice that the ranges of values for hydrophobicity, charge and aromaticity mostly overlap between WJS antibodies and in-vitro functional measurements (env5). These results reconfirm what was already included in the original WJS publication Frey et al. (2023a). In Figure 8 we investigate the humanness of the proposed antibodies. Note that since this platform requires uploading of sequences, we were not in position to score our internal experimental sequences. Hence, we provide the results only for env 0-3. These results confirm that the WJS antibodies provide a well-balanced mix of antibodies both close and far to the therapeutic human reference dataset (cf. Figure 3 in (Prihoda et al., 2022)), as expected within a drug design pipeline which leverages immunization campaigns (lead molecules from animal germlines) and ML generative models.

With this analysis in place, we are confident that our antibody benchmark is indeed representative of what we can expect in a real-world drug design framework.

## B.1 $\Delta\Delta G$ LABELS AS PROXY FOR AFFINITY MEASUREMENTS

The change in Gibbs free energy, $\Delta G$, and the dissociation constant, $K_D$, can be shown to be theoretically equivalent up to a proportionality; we have $\Delta G = RT \ln K_D$, where $R$ is the gas

Table 4: Model selection: train-domain validation set when HER2 is included.

| Algorithm | env-0 | env-1 | env-2 | env-3 | env-4 | Avg |
|---|---|---|---|---|---|---|
| | | | **SeqCNN** | | | |
| ERM | $63.2 \pm 1.0$ | $66.2 \pm 0.9$ | $66.2 \pm 0.8$ | $64.9 \pm 0.1$ | 46.6 +/- 3.5 | 61.42 |
| ERM-ENS | 62.3 | 66.1 | 69.7 | 65.9 | 51.93 | 63.18 |
| SMA | $61.8 \pm 0.9$ | $66.5 \pm 0.3$ | $66.1 \pm 0.2$ | $64.9 \pm 0.3$ | 51.9 +/- 1.5 | 62.24 |
| SMA -ENS | 58.6 | 66.9 | 68.8 | 66.1 | **55.71** | **63.22** |
| IRM | $60.0 \pm 0.9$ | $64.4 \pm 0.2$ | $69.6 \pm 0.9$ | $63.5 \pm 0.6$ | 39.1 +/- 4.4 | 59.32 |
| IRM-ENS | 62.4 | 66.5 | 73.2 | 65.1 | 43.06 | 59.32 |
| VREx | $60.1 \pm 1.6$ | $65.7 \pm 1.0$ | $66.3 \pm 0.6$ | $64.9 \pm 0.4$ | 49.7 +/- 1.8 | 61.34 |
| VREx -ENS | 61.5 | 66.9 | 68.2 | 66.1 | 48.85 | 62.31 |
| Fish | $58.2 \pm 1.3$ | $66.0 \pm 0.2$ | $68.2 \pm 0.6$ | $65.8 \pm 0.3$ | $51.5 \pm 2.2$ | 61.94 |

Table 5: Model selection: test-domain validation set (oracle) when HER2 is included.

| Algorithm | env-0 | env-1 | env-2 | env-3 | env-4 | Avg |
|---|---|---|---|---|---|---|
| | | | **SeqCNN** | | | |
| ERM | $61.4 \pm 1.4$ | $64.4 \pm 0.1$ | $66.5 \pm 0.6$ | $63.8 \pm 0.5$ | 54.5 +/- 1.1 | 62.12 |
| ERM-ENS | 62.6 | 66.2 | 68.0 | 65.9 | 38.41 | 60.22 |
| SMA | $57.0 \pm 0.4$ | $65.9 \pm 0.3$ | $65.8 \pm 0.2$ | $64.5 \pm 0.4$ | 50.7 +/- 1.7 | 60.78 |
| SMA-ENS | 61.2 | 67.0 | 67.1 | 66.2 | 45.07 | 61.34 |
| IRM | $61.7 \pm 1.4$ | $65.8 \pm 0.5$ | $68.6 \pm 1.6$ | $63.7 \pm 0.8$ | 58.9 +/- 2.3 | 63.74 |
| IRM-ENS | 62.70 | 64.34 | 63.51 | 71.56 | 59.53 | 64.32 |
| VREx | $58.4 \pm 1.4$ | $65.3 \pm 0.3$ | $65.1 \pm 0.3$ | $63.8 \pm 0.4$ | 53.6 +/- 0.1 | 61.24 |
| VREx-ENS | 60.9 | 66.8 | 67.83 | 66.73 | 40.47 | 60.54 |
| Fish | $59.9 \pm 2.6$ | $67.0 \pm 0.1$ | $69.4 \pm 0.4$ | $65.3 \pm 0.4$ | 52.0 +/- 1.2 | 62.72 |

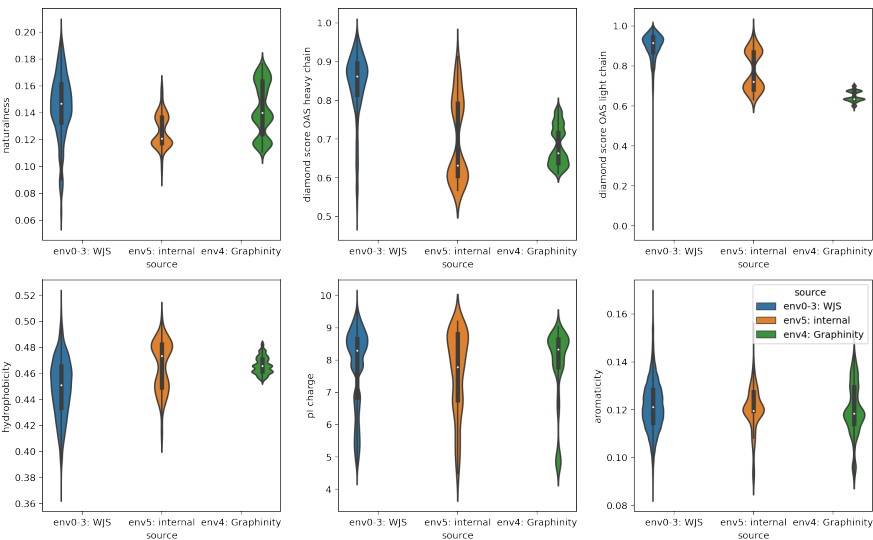

Figure 9: Various properties for evaluating quality of synthetic antibodies. Please see text for details.

constant, $1.98722\text{cal}/K \cdot \text{mol}$, and $T$ is the absolute temperature (Jorgensen & Thomas, 2008). Binding free energies have been applied to identify mutant antibodies with high affinity (Clark et al., 2017; 2019; Zhu et al., 2022), supporting the use of $\Delta\Delta G$ as a synthetic proxy for $K_D$.

That said, free energy is not exactly computable and we have used Rosetta and FoldX scores as weak approximations. The distribution of Rosetta-computed $\Delta\Delta G$ is still well-separated for binders and non-binders from fluorescence-activated cell sorting (FACS) and the separation signal is even stronger for binders from surface plasmon resonance (SPR) (Mason et al., 2021; Mahajan et al., 2022). (FACS is a higher-throughput but noisier method of identifying binders compared to SPR.)

In the case of environments 0 - 3, there is significant noise in the computed $\Delta\Delta G$ between -1 and 1 kcal/mol, because Rosetta and FoldX are less accurate at predicting $\Delta\Delta G$ for mutations with only a small effect on binding (Sirin et al., 2016; Hummer et al., 2023). We therefore remove highly uncertain labels between -0.1 and 0.1 kcal/mol before attaching binary labels: label 1 if $\Delta\Delta G < -0.1$ (stabilizing) and label 0 if $\Delta\Delta G > 0.1$ (destabilizing). We follow the following sign convention:

$$\Delta\Delta G = \Delta G_{\text{wild type}} - \Delta G_{\text{mutant}}, \tag{3}$$

such that negative $\Delta\Delta G$ is stabilizing. Note that this represents a sign flip relative to the convention followed by Hummer et al. (2023).

To validate the benefit of $\Delta\Delta G$ labels for predicting experimental binding measurements, we introduce environment 5, with details and description in Table 1. This environment consists solely of antibodies targeting a variant of the HER2 antigen. Binding labels were obtained from internally conducted surface plasmon resonance (SPR) experiments.

These variants overlap partially with antigen sequences in environment 4 (overlap between 30 - 70%); environment 4 is a subset of Graphinity (Hummer et al., 2023) and consists of synthetic single point designs aimed at HER2. The antibody sequences did not overlap with environment 4.

We report two baselines for evaluating the usefulness of training on $\Delta\Delta G$ labels:

- a random classifier.
- a binding affinity classifier, trained on 5K pairs of antibodies and binding measurements for 4 internal targets (none of them HER2). This baseline mimics the vanilla setup in drug discovery, where predictive models are confronted to a zero-shot setting where they are evaluated on new targets that differ from previous in-vitro experiments. This binding affinity classifier has accuracy of 0.64, precision 0.49 and recall 0.5.

We then run Antibody Domain bed as previously, training on env 0-4 but now evaluating on env 5. Our results are included in Table 6 and 7. We highlight two main results:

- All algorithms, including ERM achieve better results than the two baselines (binding classifier and random). This reconfirms the benefit of $\Delta\Delta G$ labels.
- CORAL, and CORAL-ENS achieve highest ac curacy, confirming the benefit of leveraging the DG algorithms and our Antibody Domainbed benchmark.

## B.2 SEQUENCE DISTANCE ACROSS ENVIRONMENTS IN ANTIBODY DOMAINBED

In what follows we compare the different environments in terms of sequence similarity. A common unit for comparison in the antibody design space is the edits, or sequence distance which is a discrete value representing number of positions with different amino-acids between two or more antibody (protein) sequences.

As we tried to separate the effect of the different generative models, by placing their corresponding designs into different environments, such split also amounts to gradually increasing the sequence distance to the seeds as the environment number progresses. From Figure 11 we notice that highest sigma environment (env 2, WJS $\sigma = 2$) include the smaller sequence distance environments (env 0 and 1, WJS $\sigma \in \{0.5, 1, 1.5\}$).

Intuitively, smaller distances between sequences should amount to similar properties, however such intuition has never been confirmed fully as there is always a counter example where even a single

Table 6: Model selection: Train-domain validation set for env 5, $K_d$ measurements prediction.

| SeqCNN | Accuracy | Precision | Recall |
|---|---|---|---|
| random classifier | 70.0 | 84.0 | 50.1 |
| binding affinity classifier | 64.0 | 49.0 | 50.0 |
| ERM | $76.0 \pm 5.7$ | $\mathbf{99.7 \pm 0.3}$ | $81.4 \pm 6.4$ |
| ERM-ENS | 72.6 | 98.31 | **91.37** |
| SMA | $61.7 \pm 8.6$ | $99.2 \pm 0.0$ | $85.0 \pm 7.5$ |
| SMA-ENS | 76.92 | 98.49 | 76.86 |
| VREx | $68.3 \pm 5.8$ | $98.9 \pm 0.2$ | $72.6 \pm 12.4$ |
| VREx-ENS | 60.00 | 99.28 | 54.12 |
| CORAL | $81.1 \pm 8.0$ | $99.4 \pm 0.0$ | $76.2 \pm 10.2$ |
| CORAL-ENS | **88.08** | 98.71 | 89.80 |

Table 7: Model selection: test-domain validation set for env 5, $K_d$ measurements prediction.

| SeqCNN | Accuracy | Precision | Recall |
|---|---|---|---|
| random classifier | 70.0 | 84.0 | 50.1 |
| binding affinity classifier | 64.0 | 49.0 | 50.0 |
| ERM | $88.1 \pm 1.8$ | $98.0 \pm 0.5$ | $88.9 \pm 1.8$ |
| ERM-ENS | 94.23 | 98.22 | 86.67 |
| SMA | $88.0 \pm 1.6$ | $99.1 \pm 0.4$ | $89.4 \pm 1.5$ |
| SMA-ENS | 94.23 | 98.35 | 93.33 |
| VREx | $89.1 \pm 2.6$ | $97.7 \pm 0.7$ | $90.2 \pm 2.4$ |
| VREx-ENS | 93.85 | 98.73 | 91.76 |
| CORAL | $91.0 \pm 3.5$ | $97.7 \pm 1.0$ | $89.7 \pm 2.1$ |
| CORAL-ENS | 82.69 | 98.66 | 86.67 |

point mutation may destroy some property of the antibody, depending on where its positioning in the sequence as well as its interaction with other atoms in the molecule or its' surroundings. We also notice this paradox in our results, the smallest distance environments usually being the most challenging one for all DG models, regardless of the backbone or the fact that such sequences have also been generated by the generative models in the other environments.

## B.3 MODEL SIZE ROBUSTNESS

We additionally explored the necessity for larger models, by fine-tuning a 4x larger ESM model with 35M parameters. Due to memory issues, we had to reduce the batch size to 8 (while ESM 8M was run with batch size 12), and to compensate we increased the number of steps to 30 000 (ESM 8M had 15 000 steps). We repeated this experiment 3 times for 5 combinations of hyper parameters. Due to the computational intensity of this model and time constraints, we could only include three of the baseline in our current results. We don't notice any gain in the performance for Antibody Domainbed by increasing the model size.

## C   CONTEXT - ANTIBODY STRUCTURE

Antibodies or immunoglobulins (Ig) maintain a common four -piece structure consisted of two identical heavy chains (HCs) and two identical light chains (LCs). The subunites are connected by disulfide bridges linking HCs and LCs together, forming the canonical "Y" shape of antibodies.

The most important regions for antibody design are the variable domains (V) of both the heavy and light chains (VH and VL, respectively). These are the regions that interact with the antigens. These domains determine the specificity of an antibody (how likely it is to attach to other molecules in the

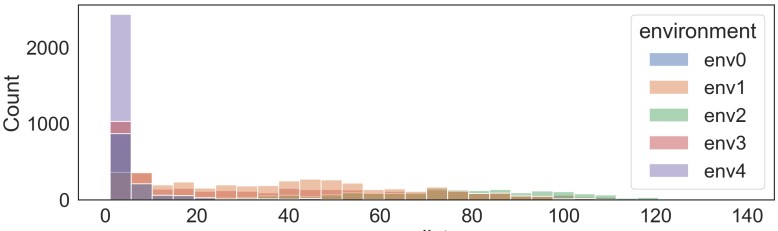

Figure 10: Sequence distances between antibody designs and their corresponding seeds. Colored by environment according to the split presented in Table 1.

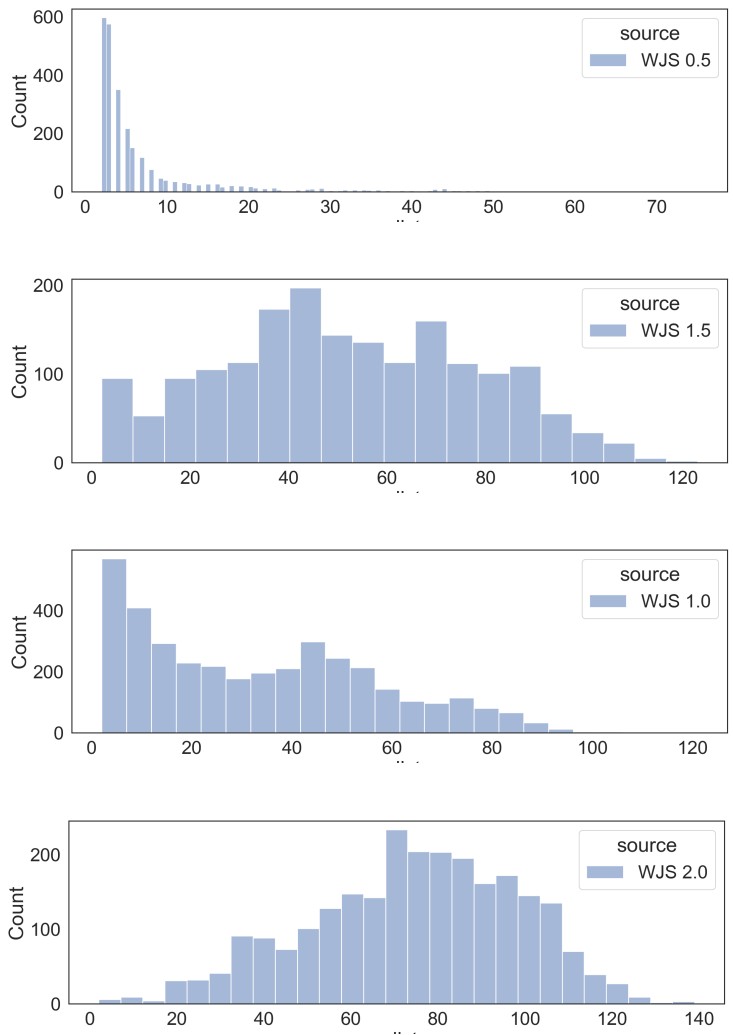

Figure 11: Sequence distances between antibody designs and their corresponding seeds for each WJS generative model.

Table 8: ESM 35M: Model selection with train and test-domain validation set.

| Algorithm | env0 | env1 | env2 | env3 | Average |
|---|---|---|---|---|---|
| | | ESM 35M: train-domain | | | |
| ERM | $61.3 \pm 1.7$ | $66.5 \pm 0.5$ | $63.8 \pm 3.7$ | $64.5 \pm 0.6$ | 64.0 |
| SMA | $62.6 \pm 1.6$ | $66.8 \pm 0.8$ | $70.5 \pm 0.0$ | $63.1 \pm 1.5$ | 65.7 |
| VREx | $61.9 \pm 0.8$ | $66.5 \pm 0.0$ | $65.9 \pm 3.4$ | $66.0 \pm 0.2$ | 65.1 |
| VREx-ENS | 62.39 | 68.27 | 71.10 | 67.49 | 67.31 |
| | | ESM 35M: test-domain | | | |
| ERM | $60.6 \pm 0.7$ | $66.8 \pm 0.7$ | $60.0 \pm 6.1$ | $63.9 \pm 0.4$ | 62.8 |
| SMA | $62.8 \pm 1.5$ | $66.8 \pm 1.2$ | $69.2 \pm 0.0$ | $63.8 \pm 2.0$ | 65.6 |
| VREx | $61.3 \pm 0.8$ | $66.7 \pm 0.5$ | $64.0 \pm 4.4$ | $65.8 \pm 0.3$ | 64.4 |
| VREx-ENS | 60.78 | 68.34 | 71.27 | 67.14 | 66.9 |

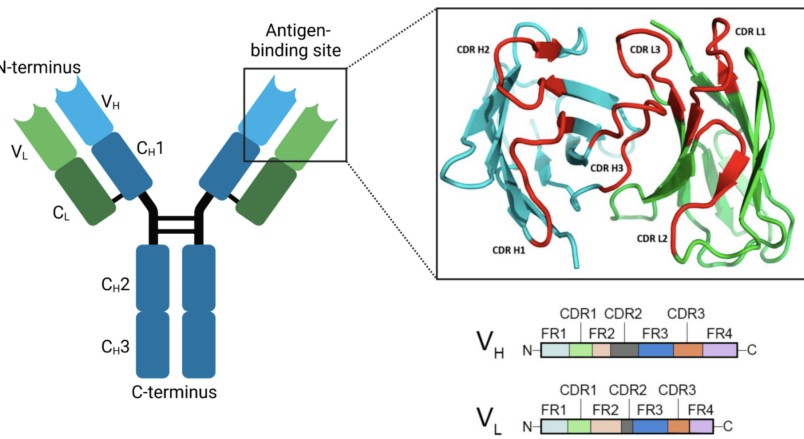

Figure 12: The antigen binding site of an antibody contains CDR and FR regions. CDR regions (L1-3 and H1-3) make up the antigen binding site on the N-terminus of the antibody.

body) through highly variable amino acid sequences. On the other hand, the constant domains (C) on heavy and light chains interact with effector proteins and molecules Figure 12.

On a more granlular level, in the VH and VL domains, there are three complementarity-determining regions: CDR-H1, CDR-H2, and CDR-H3 for VH and CDR-L1, CDR-L2, CDR-L3 for VL. These subregions are highly variable in their amino acid sequences, and they form distinct loops creating a surface complementary to distinct antigens. CDR-H3 is known to be the main contributor to antigen recognition due to its sequence diversity, length, and favourable location. Since CDR-H3 loop has an impact on the loop conformations and antigen binding at the other CDRs, it is the main driver of specificity and binding affinity. In-between the CDRs, we see the framework regions (FR). Frameworks entail less variability and provide structural stability for the whole domain. The FR regions induce a $\beta$ sheet structure, at which the CDR loops are located at the outer edge, forming an antigen-binding site.

