# OpenReview forum: "Antibody DomainBed: Out-of-Distribution Generalization in Therapeutic Protein Design"
_ICLR.cc/2024/Conference — Submitted to ICLR 2024_

### Official Review · Reviewer_keC4 · 2023-10-17

**Soundness:** 2 fair
**Presentation:** 2 fair
**Contribution:** 3 good
**Rating:** 5
**Confidence:** 3

**Summary:**

The authors identify the out-of-distribution problem in designing antibodies across multiple rounds. They propose a benchmark to mimic the distribution shift across different rounds with annotation from computational tools. Several domain generation methods as well as model ensembling in the literature of computer vision are adapted and tested in the scope of drug design on the proposed benchmark. The results suggest ensembling leads to better generalization ability.

**Strengths:**

1. Optimizing antibodies across multiple rounds is common in real-world applications, and the out-of-distribution problem is also a disturbing phenomenon in practical. Therefore, it is of great significance to formalize and put forward this problem with a suitable benchmark.
2. The authors explore many domain-generalization methods from computer vision, ensemble, as well as different model architecture on the proposed benchmark, which provides necessary basic understanding of the OOD problem in the scope of drug design.

**Weaknesses:**

1. The biggest concern is the reliability of the computational tools for annotating the binding affinity (i.e. Rosetta). As this work is proposing a benchmark on an important problem which may greatly affect the direction of the community, it is necessary to prove the fidelity of the synthetic labels. More analysis should be incorporated to prove that the computed labels are well aligned with wet-lab validations. For example, the authors can validate the accuracy of the synthetic labels on a dataset with known binding affinity and with a similar distribution to the generated sequences used in the benchmark. However, the best and most direct way might still be sampling a small subset in the benchmark and validate them with wet-lab experiments.
2. Adequate interpretation on the experimental results with respect to the proposed hypothesis is missing. In section 3.1, the authors propose the hypothesis that the causal features might be physico-chemical and geometric properties at the binding interface. However, no further analysis or interpretations on whether the tested DG algorithms have learnt this invariance are provided. Also, in Table 2 and Table 3, vanilla ERM beats most DG methods. Does this mean most of the DG methods can not learn the hypothesized invariance at all?

**Questions:**

1. In section 6.1, the claim that increasing model size improves performance is not well supported by experiments. As the experiments only compares SeqCNN with ESM2 but not models of the same type with different sizes.
2. Is ENS the abbreviation for "ensemble" in Table 2 and Table 3? And what is the "functional ensemble" mentioned in section 6.1?

---

> ### Author Response · Authors · 2023-11-22
>
> We thank the reviewer for the thoughtful review. We address the concerns below and complement it with a revision of our paper.
>
> > [ prove the fidelity of the synthetic labels. More analysis should be incorporated to prove that the computed labels are well aligned with wet-lab validations.]
>
> The change in Gibbs free energy, $\Delta G$, and the dissociation constant, $K_D$, can be shown to be theoretically equivalent up to a proportionality; we have $\Delta G = RT \ln K_D$, where $R$ is the gas constant, $1.98722 {\rm cal}/K\cdot {\rm mol}$, and $T$ is the absolute temperature [Jorgensen2008]. Free energy perturbation (FEP) has been applied to identify mutant antibodies with high affinity [Clark2017, Clark2019, Zhu2022], supporting the use of $\Delta \Delta G$ as a synthetic proxy for $K_D$.
>
> That said, free energy is not directly computable and we have used Rosetta and FoldX scores as approximations. The Rosetta-computed $\Delta \Delta G$ is still capable of separating binders and non-binders from fluorescence-activated cell sorting (FACS) well and those from surface plasmon resonance (SPR) even better [Mason2021, Mahajan2022]. (FACS is a higher-throughput, lower-SNR method of identifying binders.) These points have been added to Section A.6 of the paper.
>
> > [validate them with wet-lab experiments.]
>
> The validation on an experimental binding dataset was done in [Hummer2023], from which env 4 was curated. Models trained on FoldX-generated $\Delta \Delta G$ values are shown to achieve ROC AUC of 0.88 and average precision of 0.77 on the task of classifying binders from FACS [Mason2011].
>
> > [the causal features might be physico-chemical and geometric properties at the binding interface. However, no further analysis or interpretations on whether the tested DG algorithms have learnt this invariance are provided.]
>
> The included Domain Generalization algorithms in general attempt to learn invariant, latent representations. Since we cannot directly relate those latent features to observable physico-chemical properties, we turned to saliency mapping, Figure 6 in the manuscript. We notice that, relative to SMA and IRM, ERM displays muted behavior in the regions known to interact with the antigen paratope.
>
> > [vanilla ERM beats most DG methods. Does this mean most of the DG methods can not learn the hypothesized invariance at all?]
>
> Although this may be true on environments 0, 1, 2, 3, we observe that
> 1) in the zero shot setting (env 4) as presented in Table 4 in the Appendix, DG methods and ensembling improve over ERM when evaluated on a new target;
> 2) in Table 6 we notice significant advantage of some DG methods compared to ERM when using these models in a concept-shift setup, when the labeling function changes from $\Delta \Delta G$ to binding measurements (env 5).

---

> > ### Author Response · Authors · 2023-11-22
> >
> > > [Is ENS the abbreviation for "ensemble" ]
> >
> > Yes, we modified the manuscript to improve clarity.
> >
> > > [what is  "functional ensemble"]
> >
> > We tested two types of ensembling models; functional ensembles a.k.a. deep ensembles, average the outputs from multiple networks while weight-space ensembles e.g. SWA [Izmailov2018] average the parameters from different networks.
> >
> > [Izmailov2018] Izmailov, Pavel, et al. "Averaging weights leads to wider optima and better generalization." arXiv preprint arXiv:1803.05407 (2018).
> >
> > ### References
> >
> > [Clark2017] Clark, A. J. et al. Free energy perturbation calculation of relative binding free energy between broadly neutralizing antibodies and the gp120 glycoprotein of HIV-1. J. Mol. Biol. 429, 930–947 (2017).
> >
> > [Clark2019] Clark, A. J. et al. Relative binding affinity prediction of charge-changing sequence mutations with FEP in protein–protein interfaces. J. Mol. Biol. 431, 1481–1493 (2019).
> >
> > [Hummer2023] Hummer, A. M., Schneider, C., Chinery, L., & Deane, C. M. (2023). Investigating the Volume and Diversity of Data Needed for Generalizable Antibody-Antigen ΔΔG Prediction. bioRxiv, 2023-05.
> >
> > [Jorgensen2008] Jorgensen, W. L., & Thomas, L. L. (2008). Perspective on free-energy perturbation calculations for chemical equilibria. Journal of chemical theory and computation, 4(6), 869-876.
> >
> > [Mahajan2022] Mahajan, S. P., Ruffolo, J. A., Frick, R., & Gray, J. J. (2022). Hallucinating structure-conditioned antibody libraries for target-specific binders. Frontiers in immunology, 13, 999034.
> >
> > [Mason2021] Mason, D. M., Friedensohn, S., Weber, C. R., Jordi, C., Wagner, B., Meng, S. M., ... & Reddy, S. T. (2021). Optimization of therapeutic antibodies by predicting antigen specificity from antibody sequence via deep learning. Nature Biomedical Engineering, 5(6), 600-612.
> >
> > [Zhu2022] Zhu, F., Bourguet, F. A., Bennett, W. F., Lau, E. Y., Arrildt, K. T., Segelke, B. W., ... & Faissol, D. M. (2022). Large-scale application of free energy perturbation calculations for antibody design. Scientific Reports, 12(1), 12489.

---

### Official Review · Reviewer_nWam · 2023-10-31

**Soundness:** 2 fair
**Presentation:** 2 fair
**Contribution:** 2 fair
**Rating:** 5
**Confidence:** 3

**Summary:**

This paper proposes a benchmark dataset for assessing antibody binding affinity prediction under domain generalization. The paper motivates the need for algorithms in this setting with the rise of ML-guided design of antibodies (as well as small molecules and other biological targets), in which ML models are used to filter down to a set of candidates that are validated experimentally, the resulting data is used to modify the ML models, and the cycle repeats. To this end, the paper compiles a sequence of synthetic datasets where the antigen targets and the antibody candidates are shifted and the binary label of binding affinity is computed using a physics-based computational method. The paper then goes on to benchmark several domain-generalization approaches on these datasets under two different model architectures.

**Strengths:**

The paper considers an important problem at the intersection of domain generalization and drug design. The paper does a good job of motivating this problem in the context of active drug discovery. The paper also does a fairly extensive comparison of different domain generalization algorithms on their benchmark dataset.

**Weaknesses:**

There are a few issues with the present paper.

1. The main contribution of the paper is the benchmark dataset. However, based on my reading of this paper, it's unclear whether or not this actually a high-quality dataset. The labels are computed using a single physics-based computational method. There is no validation of the quality of these labels either experimentally or against other computational methods. Given this, it's unclear whether or not good performance on this dataset is actually meaningful.

2. The composition of the benchmark dataset seems to be somewhat arbitrary. The different environments have seemingly randomly chosen compositions of antigens and generative model parameters. There is no justification behind the choice of generative model parameters beyond the statement that changing this parameter varies the similarity between samples and starting seeds.

3. The motivation behind the paper is the active discovery setting. However, there is no evaluation of methods with respect to an active learning setup to demonstrate that better performance on the benchmark dataset leads to better active learning performance.

**Questions:**

The computational methods appear to produce real-valued results. Why are the results binarized? This would appear to throw out information.


Why do the environments consist of mixtures of antigens? It would seem to me that in a drug discovery setup, one would have a particular target in mind.

---

> ### Author Response · Authors · 2023-11-22
>
> We thank the reviewer for the thoughtful review. We address the concerns below and complement it with a revision of our paper.
>
>  > [validation of the quality of these labels either experimentally or against other computational methods]
>
> The score function we use for computing the $\Delta \Delta G$ predictions has been experimentally validated in the original publication [Park2015]. Our new encouraging results on env 5 with wet-lab binding measurements additionally verify the utility of these scores.
>
>  > [The composition of the benchmark dataset seems to be somewhat arbitrary. [...] There is no justification behind the choice of generative model parameters and other environment metadata]
>
> We set the environments to mimic the shift occuring between lab-in-the-loop iterations, Figure 2 in the manuscript, where the generative models change due to various improvements and the seeds and antigens are fixed.
> Arguably, there are multiple possible settings and users of our dataset can construct their dataset as they wish to measure specific domain shifts.
> The parameter in the WJS corresponds to the "smoothing scale," which controls the edit distance between the seed and the generated designs. Sequence similarity (or edit distance) is a common, widely-accepted metric used by scientists in the therapeutic antibody domain, as a way to categorize different 'subgroups' of antibodies [Miho2019]. We therefore rely on the edit distance as a metric to compare antibody sequences.
>
> > [demonstrate that better performance on the benchmark dataset leads to better active learning performance]
>
> The development of well-performing, robust predictors immediately translates to the success of an active learning pipeline, as these ML predictors are used in the active-learning selection process before sending sequences to a wet-lab.

---

> ### Author Response · Authors · 2023-11-22
>
> > [Why are the results binarized?]
>
> There is significant noise in the computed $\Delta \Delta G$ between -1 and 1 kcal/mol, because Rosetta and FoldX are less accurate at predicting $\Delta \Delta G$ for mutations with only a small effect on binding [Sirin2016, Hummer2023]. We therefore remove highly uncertain labels between -0.1 and 0.1 kcal/mol before attaching binary labels: label 1 if $\Delta\Delta G < -0.1$ (stabilizing) and  label 0 if $\Delta\Delta G > 0.1$ (destabilizing).
>
> Binarized labels also align with potential use cases in experimental binding prediction. While surface plasmon resonance (SPR) yields continuous affinity values, antibody screening is often carried out using noisier but higher-throughput measurement protocols like fluorescence-activated cell sorting (FACS) and enzyme-linked immunosorbent assay (ELISA) aimed at (binary) identification of binders.
>
> Lastly, algorithm development and benchmarking for domain generalization has been focused on classification tasks. Regression is an interesting future extension particularly for lab-in-the-loop design.
>
> > [environments consist of mixtures of antigens]
>
> While the reviewer is correct in that lead identification and optimization efforts are often based on a particular target determined in the target discovery stage, the training data for the models need not be restricted to the single target of interest. The mechanistic (causal) stabilizing interaction between an antibody and antigen can be learned across targets. Data from multiple targets can arguably offer more information about factors governing the interaction between the epitope and paratope.
>
> We would like to further clarify that a given target can have multiple "antigen variants" with varying sequences. We thus expose our model to various sequences on the antigen side for a given target.
>
> In addition, target generalization is of interest in the _de novo_ antibody design setting, where we initially do not have annotated binding data. Environment 4 emulates this by probing the ability of a model trained on HIV1 and SARS-CoV-2 data to generalize to an unseen target, HER2.
>
> [Park2015] Park, Keunwan, et al. "Control of repeat-protein curvature by computational protein design." Nature structural & molecular biology 22.2 (2015): 167-174.
>
> [Miho2019] Miho, E., Roškar, R., Greiff, V., & Reddy, S. T. (2019). Large-scale network analysis reveals the sequence space architecture of antibody repertoires. Nature communications, 10(1), 1321.
>
> [Sirin2016] Sirin, S., Apgar, J. R., Bennett, E. M., & Keating, A. E. (2016). AB‐Bind: antibody binding mutational database for computational affinity predictions. Protein Science, 25(2), 393-409.

---

### Official Review · Reviewer_PsDP · 2023-11-01

**Soundness:** 3 good
**Presentation:** 4 excellent
**Contribution:** 3 good
**Rating:** 6
**Confidence:** 4

**Summary:**

The paper discusses the important topic of Domain Generalization (DG) applied to the field of antigen discovery.

The paper conducts experiments aimed at proving that ensembling in output and weight space improve DG. There is also a comparison between a SeqCNN baseline and a "foundational" model for protein language.

While the topic of interest at the core of the paper's application is of particularly high interest and it is great that the authors release a new data set as part of the publication, I am not sure one can reach the conclusions of the authors with high confidence based solely on the evidence presented in the current set of experiments.

**Strengths:**

1) The paper does a very good job in my opinion of explaining the topic of Domain Generalization as part of protein design where multiple rounds of laboratory based evaluation are needed. This is also not dissimilar to other topics in AI where real world experiments help gather more data for instance as a model is deployed under a particular policy. The applicability goes potentially beyond the application presented here.

2) The methods presented to improve DG are not rocket science which does caveat the level of contribution here but also means that implementing the solutions presented by the authors is trivial and therefore industrial applications flow directly from the paper. This is a plus in my opinion as a practitioner.

3) The experiments are well explained and motivated.

**Weaknesses:**

1) I think that one might want to be a bit more careful when employing the term "foundational model". Some eminent professors in the field of AI have pointed out the somewhat misguiding nature of the term. In a scientific paper, it might be warranted to explain what exactly the authors mean by this term. Do we just mean here a model with good generalization, fine-tuning and zero-shot abilities? Can we give more details as to why the architecture considered here and the data set the model was trained on are a good fit. This is not self-evident IMO.

2) The authors only consider one baseline to compare ESM2 against without really motivating its design. I hate to be the reviewer asking for more baselines but, as such, the conclusion that using foundational models improves solving DG is tenuous based on the evidence given in the paper. More comparisons are needed I believe. At the very least, the level of confidence of the conclusion should be lowered and doubt made explicit.

3) The other conclusions, namely that resembling helps baseline models and that choosing a good validation set is important are really not surprising and do not constitute a substantial contribution.

4) I think that Figure 6 needs to be a bit more explained in appendix to novices in the field of protein design. But if that's not the target audience, it's fine.

**Questions:**

1) Could the authors please fix a few typos? The section title 5.3 is not separate from the text above it. Some xlabels in appendix are missing, it looks like the page layout cropped them out.

2) Could the authors adjust the confidence of their conclusion regarding the supremacy of the "foundational" model? While one can expect such a result, since we are entirely relying on experimental comparisons with baselines here, it is very hard to give credence to the claim with just a baseline which looks a bit arbitrarily chosen (I do agree that the design makes intuitive sense but there are other families of models that readers may want to see considered here such as simple RNN, attention models, and combinations of these approaches with CNNs).

---

> ### Author Response · Authors · 2023-11-22
>
> We thank the reviewer for the thoughtful review. We address the concerns below and complement it with a revision of our paper.
>
> > ["foundational model"]
>
> By foundation model, we meant a model pretrained on a large dataset, with good finetuning and zero-shot abilities. For more details, please see [Bommasani2021].
> We chose ESM-2, a BERT-style model pretrained on general proteins as this is a standard model for representation learning tasks. There are antibodies and antigens in the pretraining dataset.
>
> > [One baseline to compare ESM2 against [...] using foundational models improves solving DG is tenuous]:
>
> We added a graph neural network backbone, GearNet [Zhang2023], pretrained on AlphaFold2-predicted protein structures.
> Due to time constraint, we only ran some DG models (VREx, ERM and ENS).
> We observe that GearNet is the best performing model; one key reason for this boost in performance is that $\Delta\Delta G$ labels are directly computed from structure.
>
> > [The other conclusions, namely that ensembling helps baseline models and that choosing a good validation set is important are really not surprising and do not constitute a substantial contribution.]
>
> Ensembling outperformed all DG models on our task, as is the case on image classification [Arpit2022]. We cannot conclude that invariant models do not outperform ERM when the hyperparameter ranges, model size, and validation set are carefully controlled. This was the main conclusion from DomainBed on image-based datasets [Gulrajani2007]. We recover this result in the real-world application of antibody design.
> Additionally, we now update the benchmark with results on unseen target (env 4) and real binding measurements (env 5) where we observe a performance boost with ensembling and DG approaches.
> Note that for new application domains and benchmarks, where there is no standard validation set, defining a validation close to the test set is key to controlling domain shifts; this is why we believe that emphasizing the importance of choosing a good validation set can help better perform model selection.
>
> [Bommasani2021] Bommasani, Rishi, et al. "On the opportunities and risks of foundation models." arXiv preprint arXiv:2108.07258 (2021).
>
> [Arpit2022] Arpit, Devansh, et al. "Ensemble of averages: Improving model selection and boosting performance in domain generalization." Advances in Neural Information Processing Systems 35 (2022): 8265-8277.
>
> [Zhang2023] Protein Representation Learning by Geometric Structure Pretraining, ICLR 2023.
>
> [Gulrajani2007] Gulrajani, I., & Lopez-Paz, D. (2020). In search of lost domain generalization. arXiv preprint arXiv:2007.01434.

---

### Official Review · Reviewer_6GLG · 2023-11-01

**Soundness:** 3 good
**Presentation:** 3 good
**Contribution:** 3 good
**Rating:** 6
**Confidence:** 3

**Summary:**

The paper adds a new type of dataset to the DomainBed benchmark.  The dataset relates to drug discovery, an important area of study and could in principle complement DomainBed.  Despite the reasonably clear presentation of the algorithm for the construction of the data, the characterization and quality of the dataset, as well as the ability to select models and generalize from it in a drug discovery setting, remain unclear.

**Strengths:**

The paper adds a new type of dataset to the DomainBed benchmark.  The dataset relates to drug discovery, an important area of study and could in principle complement DomainBed.  Despite the reasonably clear presentation of the algorithm for the construction of the data, the characterization and quality of the dataset, as well as the ability to select models and generalize from it in a drug discovery setting, remain unclear.

**Weaknesses:**

A main weakness of this contribution is the limited nature of this one additional dataset. The dataset is fully synthetic, which the authors explain as necessary due to the high cost of antibody tests in a laboratory setting (and, I imagine, their inability to disclose actual laboratory datasets from ongoing or past projects due to competitive risks).  Although one would hope that the particular distribution shift captures an important mode of the distribution shifts observed during laboratory antibody design, it is not clear if the selection process used here ends up being educational for applications in real world settings.  Perhaps the authors have anecdotal or unpublished evidence supporting such a claim, however, this conclusion is not clear from the writing itself. It does not help that the presented characterization has jargon specialized to the field and terms like MMD (maximal mean discrepancy?) and saliency visualization that are not clarified and have scales that are illegible in the axis labels of Fig 4 and 6.

Although I don't mind people selling their own work as well as they can, I do take issue with one particular statement in page 5 that reads: "Being much larger than small molecules, they are arguably more complex and more challenging to characterize."  This statement is misleading.  Empirically, antibody design in drug discovery settings is dramatically easier than small molecule design as evidenced by the shorter average timescale of the research pipelines prior to reaching development candidates, despite the potentially much more convenient modality of a small molecule therapy if it was comparably complex to design.  The pipelines for dealing with antibodies are well established and antibody production is almost like a printing press compared to those of small molecules, for which some particular designs might take months to years to get right. I agree, however, with the sentiment of the paragraph, namely proteins present unique design challenges and it is thus useful to have an additional benchmark.

**Questions:**

Do the authors have unpublished evidence of the usefulness of the models selected by this approach in actual drug discovery settings compared to a baseline ERM approach carefully implemented?

How robust would the conclusions be under sampling of much larger computational datasets (especially more antigens but same process, but also same antigens and more designs)?  The specific construction of the inputs to the different environments basically differ in the seed sequences used in the sampling to generate new sequences that would progress to the next level.  Could it be that the authors always create a perfect setting for ensemble models and simple moving averages?  If showing this conclusion was the main purpose of the paper, are there simpler or additional datasets that could support it?

The particular 8M-parameter ESM2 model that the authors used sounds incredibly small.  It is not clear if that is a necessity (and if so, why), or if it turns out to somehow be sufficient and as good as the next level ESM2 model.

The authors (I believe) presented a closely related work at an ICML symposium this summer.  Was the main change since that presentation the inclusion of additional analysis methods, including the ensembling, or was the dataset itself improved or modified and if so how?

---

> ### Author Response · Authors · 2023-11-22
>
> We thank the reviewer for the thoughtful review. We address the concerns below and complement it with a revision of our paper.
>
> > [the construction of the data, the characterization and quality of the dataset, as well as the ability to select models and generalize from it in a drug discovery setting, remain unclear.]
>
> Please see our updated Section A.5, where we evaluate our dataset with regards to six antibody properties:
> * naturalness and diamond score on both heavy and light chains (Figure 9) which measure the closeness of an antibody to OAS; the largest publicly available human antibody database.
> * hydrophobicity, charge and aromaticity from the design sequences (Figure 9), antibody properties which contribute to binding.
> * humanness (Figure 8), leveraging the scoring framework OASis provided by BioPhi; we achieve ranges expected for a synthetic design library.
>
> Our analysis of these scores confirms that the synthetic designs in Antibody DomainBed conform with properties of natural antibodies, similarly to a point extensively explored in the original publication of the generative model we use (Frey et al. 2023).
>
> Regarding the model selection and generalization, our objective was to
> 1) explore the ability of domain generalization algorithms to deliver as expected in biological datasets and
> 2) determine whether the results are sufficient and encourage further developments of DG algorithms that take into account drug discovery pipelines.
>
> We observe encouraging results for generalizing to actual binding measurements as pointed in our response to all reviewers. We hope that by providing a complete simulation and evaluation framework we bring the two communities closer together which may initiate new progress to the benefit of both fields.
>
> > [ it is not clear if the selection process used here ends up being educational for applications in real world settings.] [Do the authors have unpublished evidence of the usefulness of the models selected by this approach in actual drug discovery settings compared to a baseline ERM approach carefully implemented?]
>
> To verify the usefulness of $\Delta\Delta G$ predictions for better binding affinity prediction, we include a new environment (env 5), consisting of real wet-lab measurements for the HER2 target.
> A seqCNN classifier trained on wet-lab measurements data on other 4 internal targets (unrelated to HER2) achieves an accuracy of 64.0\%, precision 49.0\% and recall 51.0\% which reconfirms the fundamental conundrum of drug design, namely, the ability to predict behaviour of new therapeutics on unseen targets.
> Moreover, training on $\Delta\Delta G$ predictions included in env 0-4 proves to be helpful in predicting the binding affinity for the new env 5.
> Namely, pure ERM training on $\Delta\Delta G$ scores, achieves accuracy: 76.0 +/- 5.7\%, precision: 99.7 +/- 0.3\%, recall: 83.2 +/- 4.9\% on environment 5, and the DG algorithms follow, with CORAL acheiving up to 88\% accuracy after ensembling.
> Please see Table 6 and 7 in the revision for a detailed overview of the results.
>
> > [It does not help that the presented characterization has jargon specialized to the field and terms like MMD (maximal mean discrepancy?)]
>
> We detailed in our revised manuscript the meaning and interpretation of MMD (maximum mean discrepancy), a distance measure between distributions, and hope this will be clear to future readers.
>
>  > [saliency visualization that are not clarified and have scales that are illegible in the axis labels of Fig 4 and 6.]
>
> Appologies for the ambiguity. The x-axis in the saliency figure represents the position of an amino acid in the antibody sequence, after being aligned to a uniform numbering scheme, AHo [Honegger2001].
>
> > [antibodies vs small molecule applications]
>
> We removed this statement from the manuscript. It was not our intention to compare the pipeline processes between small vs. large molecules in early-stage research, which are affected by historical reasons, resource allocation, and the speed benefits of animal immunization in the case of antibodies. We now refer only to proteins posing unique challenges, as suggested. We also believe that with the increased interest in therapeutic antibodies in biotech, there is an increasing gap between ML and this data modality as it is still underexplored by the community.

---

> ### Author Response · Authors · 2023-11-22
>
> > [Could it be that the authors always create a perfect setting for ensemble models and simple moving averages? If showing this conclusion was the main purpose of the paper, are there simpler or additional datasets that could support it?]
>
> The choice of this benchmark was done with the motivation of reproducing the lab-in-the-loop setting. The cost of wet-lab experiments leads to a low-data regime where we focus only on a few antigens and restricted number of designs. The low-data regime is particularly difficult; in the larger data regime (assuming enough diversity in data), models should perform better.
> We did not create a specific environment split to favor the ensembling models, we tried to reproduce two common domain shifts: covariate shift (variations due to the generative models in environments 0, 1, 2, 3) and concept drift (variation in the labeling function in environment 4 and 5) and note that, in these two settings, ensembles perform better.
>
> > [The particular 8M-parameter ESM2 model that the authors used sounds incredibly small. It is not clear if that is a necessity (and if so, why), or if it turns out to somehow be sufficient and as good as the next level ESM2 model.]
>
> We do not want to measure the contribution of the backbone, but rather compare models at a fixed backbone.
> Running bigger models is extremely costly computationally, especially in our framework as we are required to run 60 experiments per algorithm.
> In the short time available for the rebuttal, we provide the results for a larger 35M ESM.
> The results included in Table 7 show that there is little benefit from increasing the model's size for our benchmark dataset.
> Indeed, we are in a low data regime and finetuning overparameterized models in this regime can lead to overfitting.
>
> > [The authors presented a closely related work at an ICML symposium this summer. Was the main change since that presentation the inclusion of additional analysis methods, including the ensembling, or was the dataset itself improved or modified and if so how?]
>
> * We moved from an internal dataset to a fully public dataset as our goal was to release the benchmark to the wider community.
> * We explored and extended the available DG benchmark to recent state-of-the-art models that rely on pretrained models and ensembling and observed that these models outperform invariant based models.
> * We also conducted a more thorough experimentation of domain shifts, including a zero-shot generalization setting (env 4), to measure the performance of our models and a downstream evaluation on binding measurements (env 5).
> * Finally, we added protein structures in the dataset and added new models: a pretrained graph neural network and a bigger, 35M-parameter ESM.
>
> [Honegger2001] Honegger, Annemarie, and Andreas PluÈckthun. "Yet another numbering scheme for immunoglobulin variable domains: an automatic modeling and analysis tool." Journal of molecular biology 309.3 (2001): 657-670.

---

### Author Response · Authors · 2023-11-22

We thank the reviewers for their thoughtful comments and suggestions that have improved our paper.
We are encouraged that they acknowledge the importance and significance of the topic of Domain Generalization for drug discovery (R.1, R.2, R.3, R.4), the benefits of releasing a public benchmark for the field (R.2, R.4) and the extensive comparison it provides (R.3).

We reply below to each reviewer individually; we also provide a revision of our paper, taking into account this feedback. We look forward to further discussing with the reviewers.

Our response contains in particular the following key points.
* **Quality of dataset** (R1, R3, R4): we provide a more detailed analysis of the designs and labels in Antibody DomainBed in Appendix B, Figures 8 and 9. We now include an analysis of naturalness and hummannes of the benchmark designs. This study confirms the soundness of our dataset.

* **New environment with wet-lab measurements** (R1, R3): in prediction experiments with wet-lab binding measurements, we show how $\Delta\Delta G$ prediction and affinity prediction are related tasks. Importantly, we observe that even though the baselines in Antibody DomainBed have never seen designs with actual binding measurements, they still achieve up to 88\% accuracy, 99% precision and 91% recall scores. Detailed results are included in Table 6 and 7 in the updated manuscript. These numbers are a SOTA result and confirm the applicability and benefits of Antibody Domain in real-world drug design and discovery.

* **Model size** (R1, R4): we evaluate the performance of a bigger ESM (35M instead of 8M) to compare the performance of different model sizes. We highlight that the objective of this work is to evaluate the effect of each strategy at *fixed* backbone and model size.
* **Other baselines** (R2). We added structure information to our benchmark and included a structure-based pretrained model, GearNet. Results are now included in Table 2 (main paper) and Table 3 (App. A.1).
*  **Takeaways** (R2): after conducting the experiments suggested by the reviewers, we now include the following additional take-away messages (i) leveraging $\Delta \Delta G$ predictions is beneficial for better performance on binding prediction, (ii) Ensembled DG models provide robust binding predictions on unseen antigen targets, essential for (de-novo) drug discovery.

---

### Meta-Review · Area_Chair_6Wwm · 2023-12-07

**Metareview:**

This paper applies DG algorithms to the problem of active drug design and proposes a new OOD dataset.  The reviewers are generally not supportive of the work. The main concern lies with the quality of the new dataset.  Here are two summarizing notes posted by reviewers after the discussions:   “My biggest concern is the reliability of the synthetic labels by computational tools, and I think my concern remains after reading the responses from the authors.”  “I still believe the paper is borderline. My main concern about the paper was the quality of the dataset that the authors have curated and are hoping will be a resource to the community. In particular, I wasn't sure how much the ML community should rely on a (relatively small) benchmark dataset whose covariates and labels are completely synthetic/computational.”

**Justification For Why Not Higher Score:**

See above

**Justification For Why Not Lower Score:**

See above

---

### Decision · Program_Chairs · 2024-01-16

Reject